**Data Availability Statement:** All relevant data are within the manuscript and its Supporting Information files.

# Impact of introducing fluorescent microscopy on hospital tuberculosis control: A before-after study at a high caseload medical center in Taiwan

Hsin-Yun Sun[1,2], Jann-Yuan Wang[2], Yee-Chun Chen[2], Po-Ren Hsueh[2,3], Yi-Hsuan Chen[1], Yu-Chung Chuang 🅾[2], Chi-Tai Fang 🅾[1,2]*, Shan-Chwen Chang[2], Jung-Der Wang 🅾[2,4,5]

1 Institute of Epidemiology and Preventive Medicine, College of Public Health, National Taiwan University, Taipei, Taiwan, 2 Department of Internal Medicine, National Taiwan University Hospital, National Taiwan University College of Medicine, Taipei, Taiwan, 3 Department of Laboratory Medicine, National Taiwan University Hospital, National Taiwan University College of Medicine, Taipei, Taiwan, 4 Departments of Internal Medicine and Occupational and Environmental Medicine, National Cheng Kung University Hospital, Tainan, Taiwan, 5 Department of Public Health, College of Medicine, National Cheng Kung University, Tainan, Taiwan

* fangct@ntu.edu.tw

## Abstract

### Background

Undiagnosed tuberculosis (TB) patients hospitalized because of comorbidities constitute a challenge to TB control in hospitals. We aimed to assess the impact of introducing highly sensitive fluorescent microscopy for examining sputum smear to replace conventional microscopy under a high TB risk setting.

### Methods

We measured the impact of switch to fluorescent microscopy on the smear detection rate of culture-confirmed pulmonary TB, timing of respiratory isolation, and total non-isolated infectious person-days in hospital at a high-caseload medical center (approximately 400 TB cases annually) in Taipei. Multivariable Cox regression was applied to adjust for effects of covariates. The effect attributable to the improved smear detection rate was determined using causal mediation analysis.

### Results

After switch to fluorescence microscopy, median non-isolated infectious duration decreased from 12.5 days to 3 days (P<0.001). Compared with conventional microscopy, fluorescence microscopy increased sputum smear detection rate by two-fold (for all patients: from 22.8% to 48.1%, P<0.001; for patients with cavitary lung lesion: from 43% to 82%, P = 0.029) and was associated with a 2-fold higher likelihood of prompt respiratory isolation (odds ratio mediated by the increase in sputum smear detection rate: 1.8, 95% CI 1.3–2.5). Total non-

**Funding:** J Wang: Department of Health, Executive Yuan, Taiwan (DOH92-HP-1801 and DOH94-HP-1801). Y Chen: Ministry of Health and Welfare, Executive Yuan, Taiwan (MOHW108-TDU-B-211-133002) and (MOHW109-TDU-B-211-114002). C Fang: The financial support provided by Infectious Diseases Research and Education Center, Ministry of Health and Welfare and National Taiwan University. The funders had no role in study design, data collection and analysis, decision to publish, or preparation of the manuscript.

**Competing interests:** The authors have declared that no competing interests exist.

isolated infectious patient-days in hospital decreased by 69% (from 4,778 patient-days per year to 1,502 patient-days per year).

## Conclusions

In a high TB caseload setting, highly sensitive rapid diagnostic tools could substantially improve timing of respiratory isolation and reduce the risk of nosocomial TB transmission.

## Introduction

Pulmonary tuberculosis (TB) is an airborne disease [1]. Unless promptly isolated, hospitalized patients with active pulmonary TB can transmit to both healthcare workers (HCWs) and other patients [2]. HCWs could have an incidence rate of active TB 2- to 20-fold higher than that in general population [3]. The discovery of hospital-acquired extensively drug-resistant TB, with an extremely high mortality among HIV-positive patients, further highlights this deadly hazard [4, 5]. Rapid isolation of hospitalized patients with pulmonary TB is the pivotal step to prevent nosocomial transmission [6]. However, undiagnosed TB patients hospitalized for treatment of comorbidities constitute a challenge to TB control within hospitals.

Clinical predictive rules had been proposed to guide the decision to implement respiratory isolation [7]. An expanded isolation policy which pre-emptively isolated all patients with possibility of TB achieved immediate isolation of >95% of patients with TB in low TB risk settings [8]. Nevertheless, the same policy would be impractical in a high TB risk setting because there would be too many patients to be pre-emptively isolated [9, 10]. For a laboratory diagnosis-based respiratory isolation policy, an important barrier is the limited sensitivity of conventional sputum smear [11, 12]. Compared with conventional microscopy, fluorescence microscopy has a superior sensitivity for detecting TB bacilli [13]. In 2011, World Health Organization (WHO) recommends switching conventional to fluorescence microscopy [14], under the condition that the switch should be carefully planned at country level with training, quality assurance, and monitoring on TB detection rate [14]. WHO also endorsed highly sensitive TB nucleic acid amplification test (TB-PCR) such as Xpert® MTB/RIF (Cepheid, Sunnyvale, CA) [15]. Point-of-care Xpert® MTB/RIF reduces all-cause 12-month mortality in patients positive for TB symptoms at the time of HIV diagnosis [16]. Nevertheless, to our knowledge, the effect of introducing these more sensitive diagnostic tools on reducing risk of nosocomial TB transmission has not been documented.

Taiwan is a developed country with an incidence of TB at the range of approximately 70 per 100,000 general population in 2001 [17]. In 2003, severe acute respiratory syndrome (SARS)-related chest radiograph screenings led to the discovery of a large nosocomial TB outbreak at a rehabilitation facility in Taipei, involving 65 cases of active TB in HCWs [18]. The index case of this outbreak was an elderly patient hospitalized for stroke rehabilitation without suspicion of TB [18]. Subsequent investigations found that the problem of delayed isolation of undiagnosed TB patients also existed in other hospitals [19, 20]. In response, Taiwan Centers for Disease Control (CDC) issued guidance on nosocomial TB control [21]. To facilitate laboratory diagnosis-based isolation, Medical Center A started to roll out auramine-rhodamine staining with fluorescence microscopy since 2006 and completed the switch by early 2010s. We aimed to assess whether switching from conventional microscopy to a more sensitive rapid diagnostic tool improves early detection and prompt isolation of hospitalized patients with undiagnosed TB.

## Methods

### Setting

Medical Center A, a leading university-affiliated general hospital having the second-highest TB caseload (approximately 400 cases annually) in Taiwan, was chosen for this study. The center in Taipei had a 2,200-bed capacity and provided both primary and tertiary referral care reimbursed by National Health Insurance (NHI). The service amount steadily increased from 2001 to 2014. There were 3,454,724 outpatient visits and 91,645 admissions in 2014, nearly 2-fold than that in 2001. Medical Center A followed the guidance on hospital respiratory isolation policies issued by Taiwan CDC [21]. Contact investigation had been expanded to all HCWs who were exposed to TB patients since 2004. Sputum smear auramine-rhodamine staining with fluorescent microscopy started in 2006. The national laboratory personnel training programs for quality assurance of acid-fast staining [21], mandated by Taiwan CDC, helped to ensure the technical proficiency and performance quality of fluorescence microscopy (see details in S1 Table) [22].

### Study design

This before-after retrospective cohort study included all hospitalized patients with culture-confirmed pulmonary TB in 25 wards/units in 2001 and those in 2014. We compared the duration from admission/arrival to respiratory isolation in 2001 (conventional microscopy with Ziehl-Neelsen staining, represented the baseline situation before 2003 SARS outbreak) with that in 2014 (after full switching to fluorescent microscopy with auramine-rhodamine staining and the quality assurance program). Cox regression was used to adjust for effects of covariates. the effect mediated by improved smear detection rate was precisely identity using causal mediation analysis. The study procedure and exemption of informed consent were reviewed and approved by Research Ethics Committee of National Taiwan University Hospital (Taipei, Taiwan).

### Study procedure

For each included TB case, medical and administrative records were reviewed to determine the infectious duration. A computerized data collection form was used to systematically collect the following information from the medical records: demographic data, sputum smear and culture results, presentations, comorbidities, reasons of hospitalization, and other relevant data, with pre-defined criteria.

### Definitions

Hospitalized patients had typical presentations of pulmonary TB if they had: (a) a prolonged cough for >3 weeks; (b) clinical suspicion of pulmonary TB based on chest radiography, such as cavitary pulmonary lesions, upper lobe diseases, or miliary lesions; or (c) already received a confirmed diagnosis of pulmonary TB by a positive sputum culture of *Mycobacterium tuberculosis*, positive acid-fast stain (AFS), or positive TB PCR, before the hospitalization. The hospitalization was considered as TB-related if the chief complaint suggested an infectious etiology or the admission was for inpatient TB treatment. The hospitalization was considered comorbidities-related when the patients was admitted for management of acute complications of non-infectious diseases, such as myocardial infarction, pulmonary edema, malignancy, or acute exacerbation of chronic lung diseases.

### Identification of TB cases

We retrospectively identified all cases of culture-confirmed pulmonary TB patients in 2001 (January 1 to December 31, 2001) and in 2014 (January 1 to December 31, 2014), using a computerized registry of Mycobacteriology Laboratory. The diagnosis was verified in each case with review of medical records.

### Time to respiratory isolation

For each included infectious TB case, the zero time was the date of admission to the hospital or the date of arrival to emergency department (ER). The end of follow-up was the date when the patients was sent to a respiratory isolation room (event), the date of discharge (from hospital or ER) before respiratory isolation can be implemented (censored), the date of completion 14-day anti-TB treatment (censored), or the date of mortality due to any cause (censored). For patients who had multiple admissions or multiple positive sputum cultures, only the admission with or following the first positive sputum culture (the index culture) was used to calculate the Kaplan-Meier estimates for time to respiratory isolation.

### Total non-isolated infectious patient-days in hospital

To estimate the total non-isolated infectious patient-days in hospital, each TB case/patient was considered infectious from 3 months prior to the first positive sputum culture unless being put in a respiratory isolation room or had already received a 14-day course of at least two *in vitro* active anti-tuberculous agents after the last positive sputum culture. For those who had multiple hospitalization or had ever been transferred between wards/units before being diagnosed with pulmonary TB or adequately treated, all hospitalizations or stay in each ward/unit were counted in the calculation of total infectious patient-days.

### Statistical analysis

Statistical analyses were performed using SPSS 21.0 (IBM, Armonk, New York, USA). Cox regression, with backward selection, was used to adjust for covariates. Causal mediation analyses were performed using proc causalmed (based on logistic regression for isolation status on day 7) in SAS 9.4 (SAS Institute, Cary, North Carolina, USA). All analyses were two-sided. *P* values less than 0.05 was considered statistically significant.

## Results

### Time to respiratory isolation: 2001 vs. 2014

In 2001 and 2014, 180 of 403 (45%) and 81 of 301 (27%) patients with culture-confirmed pulmonary TB were hospitalized, respectively (Table 1). The median non-isolated infectious duration decreased from 12.5 in 2001 to 3 days in 2014 (*P*<0.001) (Table 1). Improvement occurred over all subgroups of patients (S2 Table). Fig 1 shows Kaplan-Meier estimates for time to respiratory isolation (discharge of undiagnosed TB is counted as censored rather than as the end of infectiousness) in 2001 versus that in 2014 (median: 46 vs. 19 days, p = 0.028, log-rank test). Patients with cavitary lung lesions were more quickly isolated, while lack of typical clinical presentation and hospitalized due to comorbidities were associated with delayed respiratory isolation (Table 2). After adjusting for patient characteristics, TB patients in 2014 were more quickly isolated than those in 2001 (adjusted hazard ratio [aHR] 4.7, 95% confidence interval [CI] 2.7–8.2, *P*<0.001) (Table 2: Model 1).

**Table 1. Characteristics of hospitalized patients with culture-confirmed pulmonary tuberculosis.**

| Variables | 2001 | 2014 | *P* value |
|---|---|---|---|
| Number of patients | 180 | 81 | |
| Age (years), mean (SD) | 63.0 (20.7) | 66.1 (19.8) | 0.271 |
| Men, n (%) | 126 (70.0) | 60 (74.1) | 0.501 |
| Positive sputum smear, N (%) | 41 (22.8) | 39 (48.1) | <0.001 |
| Sputum TB-PCR performed, n/N (%) | 4/41 (9.8) | 37/39 (94.9) | <0.001 |
| Negative sputum smear, N (%) | 139 (77.2) | 42 (51.9) | <0.001 |
| Sputum TB-PCR performed, n/N (%) | 25/139 (18.0) | 8/42 (19.0) | 0.876 |
| Available sputum TB PCR data, N (%) | 29 (16.1) | 45 (55.6) | <0.001 |
| Positive TB-PCR, n/N (%) | 14/29 (48.3) | 40/45 (88.9) | <0.001 |
| Patients with cavitary pulmonary lesions, n (%) | 30 (16.8)[a] | 11 (13.6) | 0.515 |
| Positive sputum smear, % (n/N) | 43.3 (13/30) | 81.8 (9/11) | 0.029 |
| Patients with non-cavitary pulmonary lesions, n (%) | 149 (83.2)[a] | 70 (86.4) | 0.515 |
| Positive sputum smear, % (n/N) | 28 (18.8) | 30 (42.9) | <0.001 |
| Patients without typical presentations, n (%) | 119 (66.1) | 48 (59.3) | 0.286 |
| Positive sputum smear, % (n/N) | 14.3 (17/119) | 27.1 (13/48) | 0.051 |
| Hospitalization due to comorbidity, n (%) | 69 (38.3) | 31 (38.3) | 0.992 |
| Positive sputum smear, % (n/N) | 14.5 (10/69) | 25.8 (8/31) | 0.173 |
| Non-chest/ID specialty attending doctors, n (%) | 124 (68.9) | 57 (70.4) | 0.810 |
| Fluoroquinolone exposure within 6 months, n (%) | 32 (17.8) | 16 (19.8) | 0.703 |
| Underlying diseases, n (%) | | | |
| Hypertension | 52 (28.9) | 28 (34.6) | 0.357 |
| Diabetes mellitus | 39 (21.7) | 26 (32.1) | 0.071 |
| Malignancy | 45 (25.0) | 17 (21.0) | 0.481 |
| Chronic lung disease | 53 (29.4) | 6 (7.4) | <0.001 |
| Chronic kidney disease | 9 (5.0) | 3 (3.7) | 0.886 |
| Congestive heart failure | 9 (5.0) | 4 (4.9) | 0.983 |
| Liver cirrhosis | 8 (4.4) | 1 (1.2) | 0.343 |
| Transplantation | 4 (2.2) | 2 (2.5) | 0.903 |
| HIV infection | 1 (0.6) | 1 (1.2) | 0.525 |
| Immediate respiratory isolation[b], n (%) | 11 (6.1) | 23 (28.4) | <0.001 |
| Non-isolated infectious duration, median (IQR), days | 12.5 (6.8–28.3) days | 3.0 (0–8.0) days | . <0.001[c] |

P values are based on chi-square test or Fisher exact test unless specified otherwise. ID: infectious diseases; NA: not available

[a]One patient in 2001 did not have chest radiography available

[b]Immediate respiratory isolation was defined as respiratory isolation on the date of admission or arrival of the emergent room.

[c]Log-rank test

## Effect of fluorescent microscopy

Switching to auramine-rhodamine staining with fluorescent microscopy doubled the overall positive sputum smear rate from 22.8% (2001) to 48.1% (2014) (*P*<0.001), particularly in patients with non-cavitary lung lesions (18.8% to 42.9%, *P*<0.001) (Table 1). Cox regression analyses shows that a positive sputum smear was associated with an earlier respiratory isolation (aHR 3.2, 95% CI 2.1–4.9, *P*<0.001) (Table 2: Model 2). Causal mediation analyses show that the two-fold higher sputum smear detection rate of fluorescence microscopy doubled the

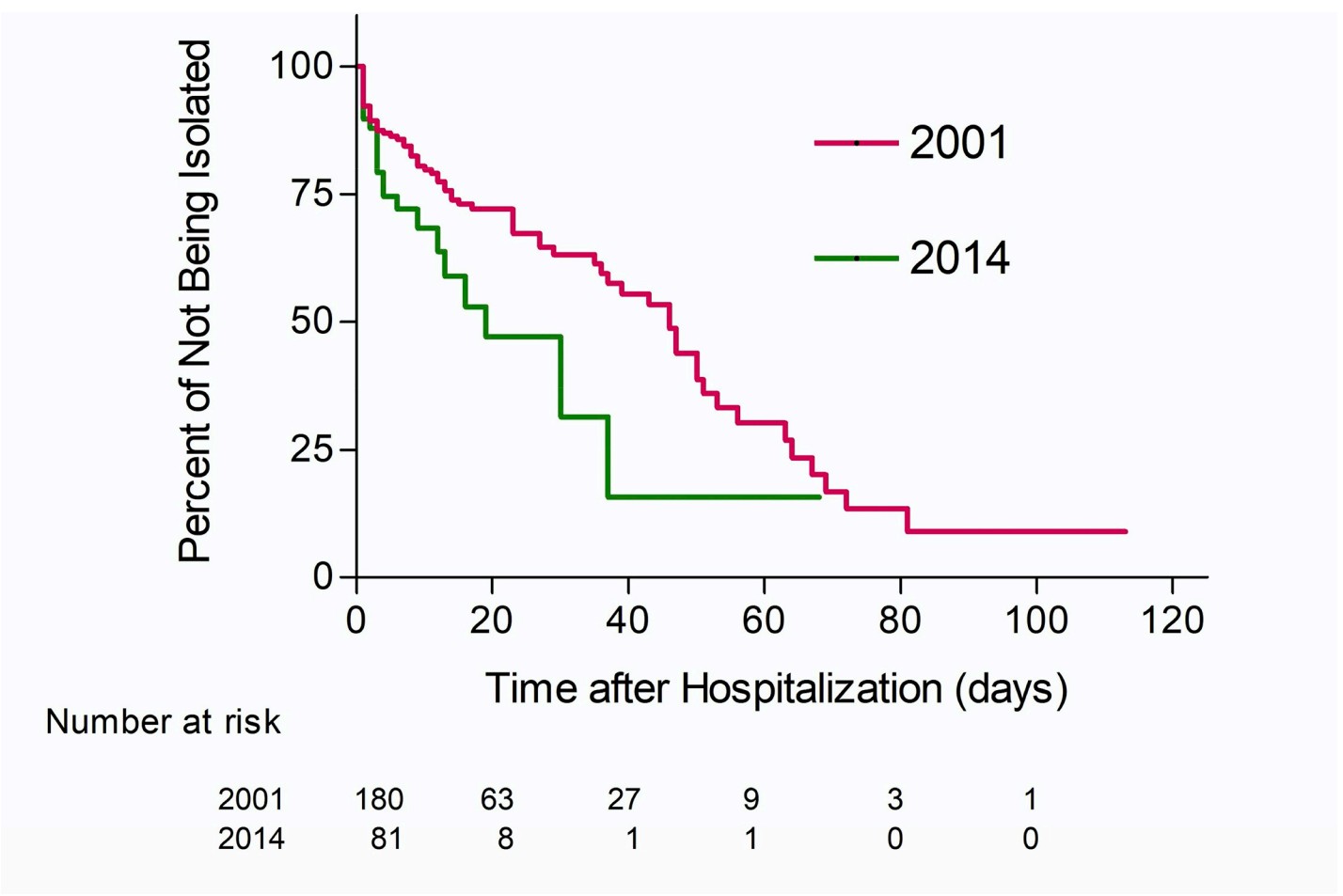

**Fig 1. Kaplan-Meier estimates for time to respiratory isolation of hospitalized patients with tuberculosis, 2001 vs. 2014.**

likelihood of early respiratory isolation (odds ratio [OR] for natural indirect effect mediated by improved sputum smear detection rate: 1.8, 95%CI 1.3–2.5, *P*<0.001) (Fig 2).

### Effect of TB-PCR

In patients with positive sputum smear, the use of TB-PCR grew from 9.8% (2001) to 94.9% (2014) (*P*<0.001); however, in patients with negative sputum smear, the use of TB-PCR remained infrequent (18.0% [2001] vs. 19.0% [2014], *P* = 0.876) (Table 1). In Cox regression analysis, TB-PCR testing was also helpful for early respiratory isolation (aHR 1.5, 95% CI 0.9–2.4, *P* = 0.094) but the effect of TB-PCR did not reach statistical significance (Table 2: Model 2).

### Physicians alertness

Alertness of physicians, measured by duration from patient arrival to physician's ordering of smear or culture, also improved from 2001 to 2014 (median: 5 vs. 2 days, *P*<0.001) (S2 Table). Cox regression analysis shows that physician alertness was also associated with earlier respiratory isolation (aHR 0.98 for each additional day before physician ordering TB smear/culture, 95% CI 0.96–0.99, *P* = 0.004) (Table 2: Model 3). Causal mediation analyses showed that improved physician alertness increased the likelihood of early respiratory isolation by 1.3-fold

**Table 2. Factors associated with prompt respiratory isolation (Model 1: 2014 vs. 2001; Model 2: The effects of positive sputum acid-fast smear and TB-PCR; Model 3: The effect of physician alertness).**

| Variables | Univariable analysis | | Multivariable analysis | | | | | |
|---|---|---|---|---|---|---|---|---|
| | HR (95% CI) | P value | aHR (95% CI)(Model 1) | P value | aHR (95% CI) (Model 2) | P value | aHR (95% CI) (Model 3)[a] | P value |
| 2014 vs. 2001 | 2.4 (1.6–3.6) | <0.001 | 4.7 (2.7–8.2) | <0.001 | 2.0 (1.2–3.4) | 0.006 | 2.7 (1.7–4.3) | <0.001 |
| Men vs. Women | 0.9 (0.6–1.3) | 0.578 | | | | | | |
| Cavitary lung lesions | 2.8 (1.9–4.1) | <0.001 | 2.0 (1.3–3.1) | 0.001 | 1.7 (1.1–2.6) | 0.022 | 1.8 (1.2–2.9) | 0.007 |
| Positive sputum smear | 5.5 (3.8–8.0) | <0.001 | | | 3.2 (2.1–4.9) | <0.001 | 3.6 (2.3–5.5) | <0.001 |
| Sputum TB-PCR test | | | | | | | | |
| Done vs. Not done | 0.3 (0.2–0.5) | <0.001 | | | 1.5 (0.9–2.4) | 0.094 | | |
| Duration from hospital visits to the date of index culture (days) | 0.96 (0.94–0.98) | <0.001 | | | | | 0.98 (0.96–0.99) | 0.004 |
| Lack of typical clinical presentations | 0.2 (0.2–0.3) | <0.001 | 0.4 (0.2–0.6) | <0.001 | 0.3 (0.2–0.5) | <0.001 | 0.3 (0.2–0.5) | <0.001 |
| Fluoroquinolone use | 0.8 (0.5–1.2) | 0.239 | | | | | | |
| Hospitalization for comorbidities | 0.4 (0.3–0.6) | <0.001 | 0.5 (0.3–0.7) | <0.001 | 0.6 (0.4–0.9) | 0.025 | | |
| Physician speciality | | | | | | | | |
| Chest/ID vs. Others | 1.4 (1.0–2.1) | 0.055 | | | | | | |
| Cancer | 0.5 (0.3–0.9) | 0.009 | | | | | | |
| Transplant recipients | 0.4 (0.1–1.5) | 0.153 | | | | | | |
| Chronic kidney disease | 1.2 (0.5–2.8) | 0.601 | | | | | | |
| Diabetes mellitus | 1.0 (0.7–1.5) | 0.952 | | | | | | |
| Chronic lung diseases | 1.2 (0.8–1.8) | 0.461 | | | | | | |
| Liver cirrhosis | 0.3 (0.1–1.2) | 0.084 | | | | | | |
| Congestive heart failure | 0.5 (0.2–1.3) | 0.177 | | | | | | |
| Hypertension | 1.1 (0.8–1.7) | 0.484 | | | | | | |
| HIV infection | 1.6 (0.2–11.4) | 0.644 | | | | | | |

HR, hazard ratio; aHR, adjusted hazard ratio

[a]Because some patients already received medical order of smear/culture at outpatient clinics, Model 3 was restricted to those who had not been suspected to have TB at admission. Only patients who had their index cultures sent after hospital visits were included for Model 3. Thus, 169 (93.9%) patients in 2001 and 73 (90.1%) in 2014 had their index cultures sent after their hospital visits (p = 0.307).

(OR for natural indirect effect mediated by early ordering of smear/culture: 1.3, 95% CI 1.02–1.5, P<0.001) (S1 Fig).

## Total non-isolated infectious patient-days per year

In 2001, there were a total of 4,778 infectious patient-days in hospital (582 from smear-positive patients, 4,196 from smear-negative patients). In 2014, the total non-isolated infectious patient-days in hospital decreased by 69%, to 1,502 infectious patient-days (229 from smear-positive patients and 1,273 from smear-negative patients). Improvement occurred over all types of wards/units, including ER, internal medicine wards, surgical wards, and intensive care units (Fig 3, S2 and S3 Figs).

## Discussion

Our study showed that introducing highly sensitive rapid diagnostic tools decreases the risk of nosocomial TB transmission from hospitalized patients with undiagnosed TB in a high TB risk setting. Switching from conventional to fluorescence microscopy doubles the sputum smear

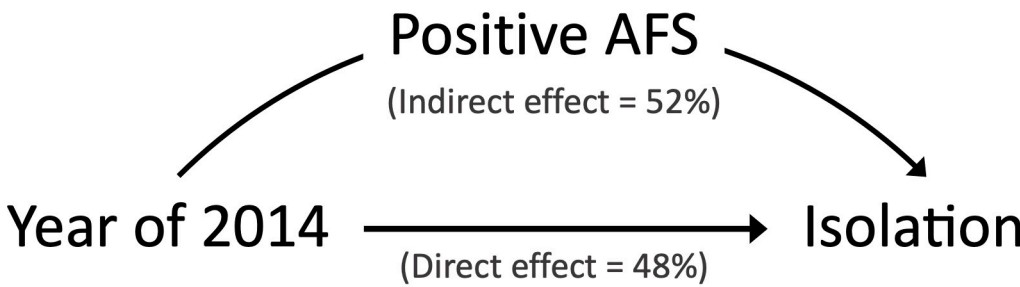

| | OR | 95% CI | p |
|---|---|---|---|
| Indirect Effect | 1.82 | (1.32-2.50) | <0.001 |
| Direct Effect | 3.80 | (1.60-9.02) | 0.002 |
| Total Effect | 6.90 | (2.78-17.10) | <0.001 |
| Percentage of Mediation | | 52.50% | |

**Fig 2. Causal mediation analyses of the effect attributable to switching from conventional to fluorescence microscopy.** Abbreviations: Year of 2014, 2014 vs. 2001; Positive AFS, positive acid-fast smear results; Isolation, respiratory isolation within 7 days.

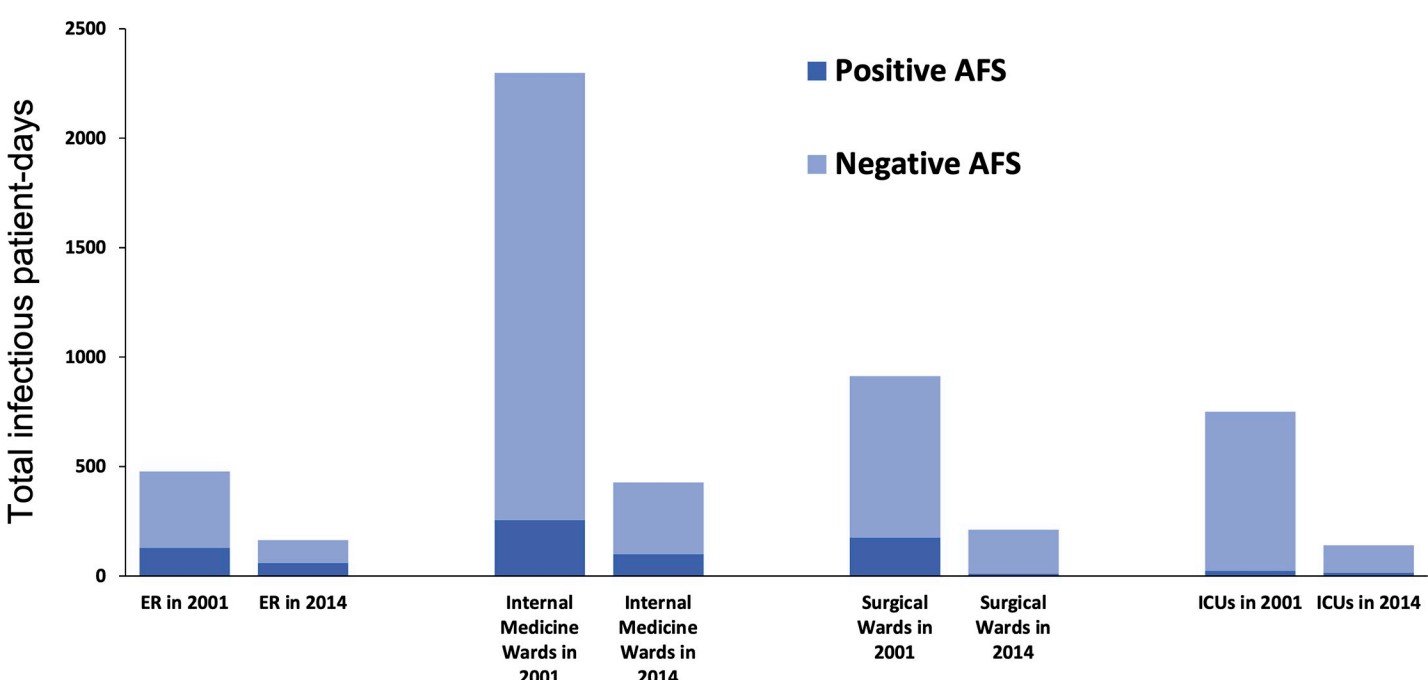

**Fig 3. Total non-isolated infectious patient-days from hospitalized patients with culture-confirmed tuberculosis, 2001 vs. 2014.** Abbreviations: AFS, acid-fast smear; ED, emergency department; ICUs, intensive care units. Internal medicine wards: general medicine, cardiovascular medicine, pulmonary medicine, endocrinology, gastro-enterology, hematology, infectious diseases, nephrology, oncology, and paediatrics. Surgical wards: cardiovascular surgery, neurosurgery, otolaryngology, general surgery, chest surgery, proctology, ophthalmology, orthopaedics, plastic surgery, and urology.

detection rate and was associated with a two-fold increase in likelihood of prompt respiratory isolation.

A genuine improvement in time-to-respiratory-isolation of hospitalized TB patients should reduce nosocomial TB transmission, especially to HWCs. This is precisely what we observed in Medical Center A. Our previous survey on age/sex-standardised TB incidence ratio of HCWs (using general population as reference)–––the excess TB risk that are attributable to nosocomial TB transmission–––in Medical Center A showed a drop of this risk, from 3.11 in 2006 to 1.37 in 2012 [23], and the decrease in time-to-isolation and total non-isolated infectious patient-days was in parallel in the present study.

Traditional Ziehl-Neelsen staining and conventional light microscopy have unsatisfactory sensitivity in detecting acid-fast bacilli [24]. A systematic review showed that, compared with conventional method, fluorescence microscopy has higher sensitivity and similar specificity [13]. This present study found that, after switch to fluorescent microscopy, the overall sputum smear detection rate doubled (23% vs. 48%, $P<0.001$), particularly in patients with non-cavitary lung lesions (18.8% to 42.9%, $P<0.001$) but also in patients with cavitary lesion (43% to 82%, $P = 0.029$). Furthermore, the superior detection rate of fluorescence microscopy translated to a more timely respiratory isolation.

The 20%–40% difference in sensitivity between fluorescence and conventional microscopy from our real life data is much larger than the 10% (on-average) reported in previous studies that compared the diagnostic performance of fluorescent versus conventional microscopy under the optimal conditions [13]. This highlights an often overlooked problem of traditional Ziehl-Neelsen staining, i.e. the majority of clinical laboratories just did not have the manpower to adequately check 300 high-power fields [13] (which takes around 4 minutes) per sample that are required for conventional microscopy [24], particularly in a busy, high clinical caseload settings. In contrast, it takes only 30–60 seconds per sample for an adequate check with fluorescent microscopy [24] which would have a decisive advantage in the real world implementation.

Although TB-PCR has been considered an important rapid diagnostic tool, this study did not show a significant role of TB-PCR in the reduction in time-to-respiratory isolation from 2001 to 2014. There were several probable reasons. First, the low adoption rate (due to cost issues) may not be enough to make an impact on shortening the overall infectious duration. Second, because of the lower sensitivity of TB-PCR in smear-negative respiratory specimens, TB-PCR was performed predominantly in specimens with positive AFS to distinguish *M. tuberculosis* from non-tuberculous mycobacteria. The negative result does not exclude the possibility that universal use of automatic TB-PCR, such as Xpert® MTB/RIF, may further shorten the time to respiratory isolation.

In keeping with previous observations (S3 Table), the absence of cough and other typical symptoms is a barrier to prompt respiratory isolation of TB patients (S2 Table). Another cause of delay is hospitalization due to comorbidities, a problem that was previously neglected or confused with the lack of typical clinical presentations. To distinguish these two different situations, we defined hospitalization due to comorbidities as the reason of admission being for the management of acute complications from non-infectious diseases, while lack of typical presentation was defined as the absence of prolonged cough for more than 3 weeks, clinical suspicion of pulmonary TB based on chest radiography, or already having a confirmed diagnosis. Multivariable regression established that hospitalization due to comorbidities was a risk factor independent from lack of typical presentations (Table 2).

The diverse distribution of hospitalized infectious TB patients across 25 medical/surgical sub-specialities wards/units (S2 and S3 Figs) further supports that hospitalization of patients with undiagnosed TB for treatment of comorbidities is an unrecognized but important issue

requiring to be addressed. The HCWs in specialties units often concentrate on the management of acute complications of non-infectious chronic diseases and are not trained or prepared for diagnosing concomitant TB in their patients. The 2003 large nosocomial TB outbreak in Taipei [18] involving more than 65 HCWs, occurred exactly under such scenario —an elderly patient with acute stroke was hospitalized to a rehabilitation unit, without being suspected to also have active TB. The harm is two-way. First, the patients are at increased risk for morbidities and mortality from delayed diagnosis and treatment of TB. Second, the HCWs are at increased risk of nosocomial TB outbreak, which could be fatal if the strain is multidrug-resistant or extensively drug-resistant [25].

The interplay between TB and chronic diseases further complicates the clinical scenarios. Chronic non-communicable diseases increase the risk of TB—the associations have been well established for diabetes mellitus [26] and rheumatoid arthritis treated with anti-tumour necrosis factor (TNF) agents [27]. On the other hand, TB may increases the risk of complications from chronic diseases, e.g. hyperglycaemia in diabetes mellitus and ischemic stroke in people with atherosclerosis [28]. Moreover, certain risk factor, such as smoking, increases the risk of both TB and chronic diseases [29]. Therefore, hospitalization of undiagnosed TB patients for treatment of comorbidities is more than just a coincidence by chance. In low HIV prevalence countries, TB is a disease of elderly [30]. With population aging, concurrence of TB and comorbidities could be an increasing challenge to clinicians.

Currently, most hospitals in Taiwan still used traditional Ziehl-Neelsen staining (or Kinyoun staining, a similar method) and conventional light microscopy in laboratory diagnosis of TB. An analysis of 2003–2010 Taiwan National Health Insurance database revealed that frequent exposure to hospital environment is a risk factor for contracting TB in Taiwan (adjusted odds ratio: 1.77, for those with $\geq$ 30 outpatient care visit annually) [31]. Our findings on the impact of switching to auramine-rhodamine staining with fluorescence microscopy suggest that a nationwide adoption and roll-out might cut the risk of nosocomial TB transmission to both patients and healthcare workers.

To address the remaining barriers to respiratory isolation of hospitalized TB patients, "FAST (Find cases Actively by cough surveillance and rapid molecular sputum testing, Separate safely, and Treat effectively based on rapid drug susceptibility test)" is an option to decrease the time to respiratory isolation [32, 33] although the substantial cost of rapid molecular testing could be a barrier to NHI reimbursement. Alternatively, environmental controls—the second level in the hierarchy of TB control—with adequate ventilation (>6–12 air changes per hour) or upper room ultraviolet germicidal irradiation (to disinfect the air) can be applied to reduce the concentration of infectious droplet nuclei in hospital indoor air, and decrease risk of nosocomial transmission [6, 34].

The present study has several important limitations. First, it was not a randomized controlled trial (which cannot be performed in the context, due to ethical reasons). Physician education and expansion of respiratory isolation facilities were simultaneously implemented along with the switching to fluorescence microscopy during the same intervention period. Nevertheless, we applied multivariable regression and causal mediation analyses to estimate the effect attributable to switching to fluorescence microscopy. Second, the study hospital closely followed the national guidance on nosocomial TB control policies issued by Taiwan CDC. Therefore, our findings might not be generalizable to hospitals with less proficiency. Third, the study hospital is a medical center, and therefore might not represent situations in smaller hospitals. Nevertheless, under NHI in Taiwan, people can and did directly seek health care in medical centers, without the need of referral from general practitioners. The 3 million annual outpatient visits and more than 90 thousand hospitalizations per year in the study hospital were predominantly from primary care services, rather than tertiary referral care. Finally,

since Taiwan has a low HIV prevalence [35], only two of TB patients had HIV coinfection in our study. Thus, our conclusions might not be generalized to countries with a high HIV prevalence.

In conclusion, highly sensitive rapid diagnostic tools could substantially improve timing of respiratory isolation and reduce risk of nosocomial tuberculosis transmission in high TB risk settings. Lack of typical presentations and hospitalization due to comorbidities continued to be main reasons of delayed isolation. Studies will be required to assess whether routine sputum smear or TB-PCR of all hospitalized patients with cough or abnormal chest radiograph is effective in overcoming these remaining barriers.

## Supporting information

**S1 Table. Tuberculosis control practices, 2001–2014.**
(DOCX)

**S2 Table. Non-isolated infectious duration of hospitalized patients (at the first admission after the index culture).**
(DOCX)

**S3 Table. Studies which investigating the factors associated with delayed respiratory isolation of hospitalized patients with pulmonary tuberculosis.**
(DOCX)

**S1 Fig. Causal mediation analyses of the effect mediated by the higher rate of early ordering (less than 4 days after admission) of smear/culture in 2014.**
(DOCX)

**S2 Fig. Number of infectious tuberculosis patients in 25 medical/surgical subspecialty wards/units in 2001 and 2014.**
(DOCX)

**S3 Fig. Total non-isolated infectious patient-days in 25 medical/surgical subspecialty wards/units in 2001 and 2014.**
(DOCX)

**S1 Data.**
(XLSX)

## Acknowledgments

We thank all staff in the hospital for their commitment to improving patient safety and reducing healthcare-associated infection.

## Author Contributions

**Conceptualization:** Chi-Tai Fang, Jung-Der Wang.

**Data curation:** Hsin-Yun Sun.

**Funding acquisition:** Yee-Chun Chen, Chi-Tai Fang, Jung-Der Wang.

**Methodology:** Jann-Yuan Wang, Po-Ren Hsueh, Yi-Hsuan Chen, Yu-Chung Chuang, Chi-Tai Fang.

**Software:** Yi-Hsuan Chen, Yu-Chung Chuang.

**Supervision:** Yee-Chun Chen, Chi-Tai Fang, Shan-Chwen Chang.

**Writing – original draft:** Hsin-Yun Sun.

**Writing – review & editing:** Hsin-Yun Sun, Jann-Yuan Wang, Yee-Chun Chen, Po-Ren Hsueh, Yi-Hsuan Chen, Yu-Chung Chuang, Chi-Tai Fang, Shan-Chwen Chang, Jung-Der Wang.

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
