## [Decision Letter · Decision Letter 0]

27 Jan 2020

PONE-D-19-32578

Impact of introducing fluorescent microscopy on hospital tuberculosis control: a before-after study at a high caseload medical center in Taiwan

PLOS ONE

Dear Dr. Fang,

Thank you for submitting your manuscript to PLOS ONE. After careful consideration, we feel that it has merit but does not fully meet PLOS ONE’s publication criteria as it currently stands. Therefore, we invite you to submit a revised version of the manuscript that addresses the points raised during the review process.

We would appreciate receiving your revised manuscript. To enhance the reproducibility of your results, we recommend that if applicable you deposit your laboratory protocols in protocols.io, where a protocol can be assigned its own identifier (DOI) such that it can be cited independently in the future. For instructions see: http://journals.plos.org/plosone/s/submission-guidelines#loc-laboratory-protocols

We look forward to receiving your revised manuscript.

Kind regards,

Frederick Quinn

Academic Editor

PLOS ONE

Journal Requirements:

Reviewers' comments:

Reviewer's Responses to Questions

**Comments to the Author**

1. Is the manuscript technically sound, and do the data support the conclusions?

Reviewer #1: Yes

Reviewer #2: Yes

2. Has the statistical analysis been performed appropriately and rigorously? 

Reviewer #1: Yes

Reviewer #2: Yes

3. Have the authors made all data underlying the findings in their manuscript fully available?

Reviewer #1: Yes

Reviewer #2: Yes

4. Is the manuscript presented in an intelligible fashion and written in standard English?

Reviewer #1: Yes

Reviewer #2: Yes

5. Review Comments to the Author

Reviewer #1: This manuscript had described the introduction of a novel fluorescent microscopy methodology in the diagnosis of TB. Globally, TB is still a major concern in public health, thus this manuscript contains importance and significance for clinicians. My suggestion is minor revision, with my comments as follows.

Firstly, the authors had mentioned the moderate to high case load several times in this manuscript, and I reckon the high cases in this manuscript is one remarkable highlight which would be more convincing and supportive. However, what is the definition of the moderate to high, or high case load in regards with TB?

Secondly, despite the large number of cases involved, this study had been performed in the Medical Center A as a single center study. Please provide some information about the routine/conventional diagnosis of TB in other medical settings and how this introduction of fluorescent microscopy methodology would improve or make difference in other hospitals in Taiwan.

Thirdly, the writing could be benefit from language editing. For example, a lot of the sentences begin with subjective tense like “we…”. Some description could be more concise, for example, the approval of ethics could be combined as one sentence. Also, “we provide the data… showing that introducing… helps to…” would be a little bit too oral, “was parallel to the decrease” should be “in parallel”.

Reviewer #2: I found this manuscript to be relevant, sound and well-written, apart from a few instances e.g.

(a) Paragraph 2 of the "Introduction" section, Page 3; Remove "Fluorescence microscopy is a more sensitive test to diagnose pulmonary TB than conventional microscopy [16]" as it is a repetition; it is mentioned in the aforementioned sentences.

(b) Still in the same section as above, the last sentence in the paragraph i.e. "Nevertheless, the effect of introducing these more sensitive diagnostic tools on risk of nosocomial TB transmission has not been studied" may not be entirely accurate as you claim. Perhaps in high-income countries with a low burden of TB (e.g. Taiwan). I suggest you revisit this.

(c) 1st paragraph on page 5, Methods section; 'strain' appears to be a typo. Did you mean 'stain'? i.e. "---already received a confirmed diagnosis of pulmonary TB by a positive sputum culture of Mycobacterium tuberculosis, positive acid-fast strain (AFS),---" should be "---already received a confirmed diagnosis of pulmonary TB by a positive sputum culture of Mycobacterium tuberculosis, positive acid-fast stain (AFS),---"

6. PLOS authors have the option to publish the peer review history of their article (what does this mean?). If published, this will include your full peer review and any attached files.

Reviewer #1: No

Reviewer #2: Yes: David Patrick Kateete

---

## [Author Response · Author response to Decision Letter 0]

12 Feb 2020

Dear Prof. Quinn,

Thank you for your encouraging response to our work. We greatly appreciate the reviewers’ constructive comments. We have endeavored to incorporate the feedback and revised our manuscript accordingly, with alterations highlighted in red color. Responses to each point of the reviewer’s comments were listed below.

Thank you again for your kind consideration and look forward to hearing from you soon.

Sincerely yours,

Chi-Tai Fang, MD, PhD

Professor

Division of Infectious Diseases

National Taiwan University Hospital

7 Chun-Shan South Road., Taipei 100, Taiwan.

Phone: +886 2 3366-8035

E-mail: fangct@ntu.edu.tw

 

To Reviewer 1:

Thank you for your positive response to our work and the kind advice. We greatly appreciate your constructive comments that have helped us improve our paper. We have endeavored to incorporate the feedback and revised our manuscript accordingly. The itemized response (abbreviated as R) are as follows:

C1. Firstly, the authors had mentioned the moderate to high case load several times in this manuscript, and I reckon the high cases in this manuscript is one remarkable highlight which would be more convincing and supportive. However, what is the definition of the moderate to high, or high case load in regards with TB?

R1. There are approximately 400 culture-confirmed TB cases annually in medical center A (ll. 33-34, 94 and 172), which has the second-highest caseload in Taiwan. We admit that we did not have a definition for “moderate to high” or “high” case load. For consistency, we revised all “moderate-to-high” to “high”. (ll. 30, 45, 62, 223, 337)

C2. Secondly, despite the large number of cases involved, this study had been performed in the Medical Center A as a single center study. Please provide some information about the routine/conventional diagnosis of TB in other medical settings and how this introduction of fluorescent microscopy methodology would improve or make difference in other hospitals in Taiwan.

R2. We added a paragraph in Discussion section to address these important issues:

“Currently, most hospitals in Taiwan still used traditional Ziehl-Neelsen staining (or Kinyoun staining, a similar method) and conventional light microscopy in laboratory diagnosis of TB. An analysis of 2003-2010 Taiwan National Health Insurance database revealed that frequent exposure to hospital environment is a risk factor for contracting TB in Taiwan (adjusted odds ratio: 1.77, for those with ≥ 30 outpatient care visit annually) [31]. Our findings on the impact of switching to auramine-rhodamine staining with fluorescence microscopy suggest that a nationwide adoption and roll-out might cut the risk of nosocomial TB transmission to both patients and healthcare workers.” (ll. 301-308)

C3. Thirdly, the writing could be benefit from language editing. For example, a lot of the sentences begin with subjective tense like “we…”. Some description could be more concise, for example, the approval of ethics could be combined as one sentence. Also, “we provide the data… showing that introducing… helps to…” would be a little bit too oral, “was parallel to the decrease” should be “in parallel”.

R3. The manuscript has been edited to avoid repetitive use of sentence begin with subjective tense “We…”. The following sentences were also revised as recommended:

 “The study procedure and exemption of informed consent were reviewed and approved by Research Ethics Committee of National Taiwan University Hospital (Taipei, Taiwan).” (ll. 114-115)

 “Our study showed that introducing highly sensitive rapid diagnostic tools decreases the risk of nosocomial TB transmission from hospitalized patients with undiagnosed TB in a high TB risk setting.” (ll. 222-224)

 “the decrease in time-to-isolation and total non-isolated infectious patient-days was in parallel in the present study.” (ll. 233-234)

 

To Reviewer #2: 

Thank you for your positive response to our work and the kind advice. We greatly appreciate your constructive comments that have helped us improve our paper. We have endeavored to incorporate the feedback and revised our manuscript accordingly. The itemized response (abbreviated as R) are as follows:

C1. I found this manuscript to be relevant, sound and well-written, apart from a few instances e.g. (a) Paragraph 2 of the "Introduction" section, Page 3; Remove "Fluorescence microscopy is a more sensitive test to diagnose pulmonary TB than conventional microscopy [16]" as it is a repetition; it is mentioned in the aforementioned sentences.

R1. We thank the reviewer for the kind comments and deleted the repetitive sentence as suggested.

C2. (b) Still in the same section as above, the last sentence in the paragraph i.e. "Nevertheless, the effect of introducing these more sensitive diagnostic tools on risk of nosocomial TB transmission has not been studied" may not be entirely accurate as you claim. Perhaps in high-income countries with a low burden of TB (e.g. Taiwan). I suggest you revisit this.

R2. We searched PubMed but did not find any published study which assessed the effect of introducing more sensitive diagnostic tools on risk of nosocomial TB transmission in either high-income or low-income countries. However, we cannot exclude the possibility that such assessment may have been performed but not published in academic journals. We therefore revised the sentences as below:

“to our knowledge, the effect of introducing these more sensitive diagnostic tools on reducing risk of nosocomial TB transmission has not been documented.” (II. 72-74)

C3. (c) 1st paragraph on page 5, Methods section; 'strain' appears to be a typo. Did you mean 'stain'? i.e. "---already received a confirmed diagnosis of pulmonary TB by a positive sputum culture of Mycobacterium tuberculosis, positive acid-fast strain (AFS),---"

R3. We apologize for the typo. The correct spelling should be:

 “positive acid-fast stain (AFS)” (ll. 129)

---

## [Decision Letter · Decision Letter 1]

21 Feb 2020

Impact of introducing fluorescent microscopy on hospital tuberculosis control: a before-after study at a high caseload medical center in Taiwan

PONE-D-19-32578R1

Dear Dr. Fang,

We are pleased to inform you that your manuscript has been judged scientifically suitable for publication and will be formally accepted for publication once it complies with all outstanding technical requirements.

With kind regards,

Frederick Quinn

Academic Editor

PLOS ONE

Additional Editor Comments (optional):

Reviewers' comments:

Reviewer's Responses to Questions

**Comments to the Author**

1. If the authors have adequately addressed your comments raised in a previous round of review and you feel that this manuscript is now acceptable for publication, you may indicate that here to bypass the “Comments to the Author” section, enter your conflict of interest statement in the “Confidential to Editor” section, and submit your "Accept" recommendation.

Reviewer #1: All comments have been addressed

Reviewer #2: All comments have been addressed

2. Is the manuscript technically sound, and do the data support the conclusions?

Reviewer #1: Yes

Reviewer #2: Yes

3. Has the statistical analysis been performed appropriately and rigorously? 

Reviewer #1: Yes

Reviewer #2: Yes

4. Have the authors made all data underlying the findings in their manuscript fully available?

Reviewer #1: Yes

Reviewer #2: Yes

5. Is the manuscript presented in an intelligible fashion and written in standard English?

Reviewer #1: Yes

Reviewer #2: (No Response)

6. Review Comments to the Author

Reviewer #1: (No Response)

Reviewer #2: Authors have satisfactorily addressed all the comments. I recommend that the revised manuscript be published.

7. PLOS authors have the option to publish the peer review history of their article (what does this mean?). If published, this will include your full peer review and any attached files.

Reviewer #1: Yes: Zhenbo Xu

Reviewer #2: Yes: David Patrick Kateete

---

## [Editor Report · Acceptance letter]

23 Mar 2020

PONE-D-19-32578R1 

Impact of introducing fluorescent microscopy on hospital tuberculosis control: a before-after study at a high caseload medical center in Taiwan 

Dear Dr. Fang:

I am pleased to inform you that your manuscript has been deemed suitable for publication in PLOS ONE. Congratulations! Your manuscript is now with our production department. 

With kind regards,

on behalf of

Dr. Frederick Quinn 

Academic Editor

PLOS ONE